# Immunity and Protective Efficacy of a Plant-Based Tobacco Mosaic Virus-like Nanoparticle Vaccine against Influenza a Virus in Mice

**DOI:** 10.3390/vaccines12101100

**Published:** 2024-09-26

**Authors:** Adthakorn Madapong, Erika M. Petro-Turnquist, Richard J. Webby, Alison A. McCormick, Eric A. Weaver

**Affiliations:** 1Nebraska Center for Virology, University of Nebraska-Lincoln, Lincoln, NE 68503, USA; amadapong2@unl.edu (A.M.); epetro-turnquist2@huskers.unl.edu (E.M.P.-T.); 2School of Biological Sciences, College of Arts and Sciences, University of Nebraska-Lincoln, Lincoln, NE 68503, USA; 3St. Jude Children’s Research Hospital, Memphis, TN 38105, USA; richard.webby@stjude.org; 4College of Pharmacy, Touro University California, Vallejo, CA 94592, USA; alimccormick@ucdavis.edu

**Keywords:** hemagglutinin, influenza A virus, mouse, plant-based manufacturing, tobacco mosaic virus, vaccine

## Abstract

Background: The rapid production of influenza vaccines is crucial to meet increasing pandemic response demands. Here, we developed plant-made vaccines comprising centralized consensus influenza hemagglutinin (HA-con) proteins (H1 and H3 subtypes) conjugated to a modified plant virus, tobacco mosaic virus (TMV) nanoparticle (TMV-HA-con). Methods: We compared immune responses and protective efficacy against historical H1 or H3 influenza A virus infections among TMV-HA-con, HA-con protein combined with AddaVax™ adjuvant, and whole-inactivated virus vaccine (Fluzone^®^). Results: Immunogenicity studies demonstrated robust IgG, IgM, and IgA responses in the TMV-HA-con and HA-con protein vaccinated groups, with relatively low induction of interferon (IFN)-γ^+^ T-cell responses across all vaccinated groups. The TMV-HA-con and HA-con protein groups displayed partial protection (100% and 80% survival) with minimal weight loss following challenge with two H1N1 strains. The HA-con protein group exhibited 80% and 100% survival against two H3 strains, whereas the TMV-HA-con groups showed reduced protection (20% survival). The Fluzone^®^ group conferred 20–100% survival against two H1N1 strains and one H3N1 strain, but did not protect against H3N2 infection. Conclusions: Our findings indicate that TMV-HA and HA-con protein vaccines with adjuvant induce protective immune responses against influenza A virus infections. Furthermore, our results underscore the potential of plant-based production using TMV-like nanoparticles for developing influenza A virus candidate vaccines.

## 1. Introduction

Influenza epidemics pose a significant global health challenge, infecting up to 1 billion people annually and resulting in 3–5 million severe cases, and up to 750,000 deaths worldwide [1]. Symptoms typically include fever, headache, myalgia, and respiratory distress lasting 5–15 days. The elderly and immunocompromised individuals are particularly vulnerable to severe morbidity and mortality from influenza infections [2,3,4]. Pandemic outbreaks, such as the 2009 H1N1 swine influenza pandemic, underscore the urgent need for effective treatments, affecting over 24% of the global population [5]. Similarly, the H3 subtype, originating from avian sources in 1968, continues to circulate and cause seasonal epidemics [6,7].

Annual influenza vaccination is recommended as the best way to protect against influenza infections due to the change in influenza virus types, subtypes, and phenotypes. The World Health Organization (WHO) and the Centers for Disease Control and Prevention (CDC) use influenza virus surveillance to predict the strains likely to circulate in the upcoming influenza season. Based on these predictions, they recommend the annual quadrivalent vaccine formulation, which includes H1N1, H3N2, and influenza B virus strains [8,9]. However, vaccine effectiveness (VE) is highly variable, ranging from 10 to 60%, and provides limited cross-protection against mismatched strains [10].

Influenza vaccine immunogens have been delivered by several mechanisms, including viral vectors. Centralized hemagglutinin (HA) genes were delivered by a range of human adenovirus vectors in previous studies [11,12,13,14]. While viral vectors are highly immunogenic, some significant adverse effects have been described [15,16]. Nanoparticle (NP)-based vaccines are an appealing platform, as they can be administered through various routes of administration, stimulate strong immune responses, and be easily tailored to address emerging diseases [17,18]. Recently, virus-like particles (VLPs) have been developed as a type of NP-based vaccine that have been shown to be efficacious for hepatitis B virus and human papillomavirus [19], respiratory syncytial virus [20], influenza [21], and other microbial diseases. Plants have a long history as a manufacturing platform for VLPs, and various plant-manufactured VLPs demonstrate promise in efficacy testing [22]. Plant viruses, particularly TMV, are effective immunogenic vehicles that have been used to present epitopes or immunological domains through conjugation or genetic fusion with their capsid proteins [23], and provide promising results in several preclinical trials [24,25].

Centralized vaccine antigens have been intensively studied for their ability to induce broader cross-protective immunity than wild-type antigens [26,27,28,29,30]. A vaccine that is matched to the circulating strains is likely to induce protective immunity to those strains. However, when the vaccine does not match the contemporary strains, a centralized immunogen can induce cross-protection against divergent viruses. In the present study, we developed an NP-based vaccine platform by conjugating recombinant protein antigens associated with the TMV virion [23]. The recombinant protein antigens represent the conserved regions of HA from either H1 or H3 strains of influenza A viruses with a commercial whole-inactivated virus vaccine Fluzone^®^ (Swiftwater, PA 18370 USA). We observed that these TMV nanoparticle vaccines induced strong antibody response and led to protection from multiple different challenge strains in the mouse model.

## 2. Materials and Methods

### 2.1. Ethics Statement

All biological procedures were reviewed and approved by the Institutional Biosafety Committee (IBC) at the University of Nebraska-Lincoln (UNL). Female BALB/c mice, aged 5–6 weeks, were purchased from Jackson Laboratory and housed in the Life Sciences Annex building on the UNL campus, following the guidelines of the Association for Assessment and Accreditation of Laboratory Animal Care International (AAALAC). The mice were kept in a Tecniplast IVC caging system with recycled paper bedding (Tek-fresh, Envigo Bioproducts, Madison, WI, USA) and provided a standard rodent chow diet (Envigo Bioproducts). Enrichment items, such as a Kimwipe, Nylabone, or plastic hut, were also included. All protocols were approved by the UNL Institutional Animal Care and Use Committee (IACUC), under project ID 2158. All animal experiments were conducted in accordance with the Animal Welfare Act, PHS Animal Welfare Policy, the NIH Guide for the Care and Use of Laboratory Animals, and the policies and procedures of UNL.

### 2.2. Influenza Viruses

The following human influenza viruses were obtained from the ATCC or the Biodefense and Emerging Infectious Diseases Repository (BEI): Influenza virus H1N1 A/Puerto Rico/8/1934 (ATCC-VR95), H1N1 A/Fort Monmouth/1/1947 (NR-3170, BEI), H3N1 A/Texas/1/1977 (NR-3604, BEI), and H3N2 A/Aichi/2/1968 (NR-3483, BEI). The mouse-adapted H1 and H3 influenza viruses were obtained through serial lung passaging in mice, as previously described [31]. All the viruses were grown in SPF embryonated chicken eggs and the chorioallantoic fluid was stored at −80 °C. The 50% mouse lethal dose (MLD_50_) of the influenza viruses were quantified in BALB/c mice and quantified based on hemagglutination units (HAU) and 50% tissue culture infective dose (TCID_50_). No select agents were used in any of these studies, and all viruses and animal procedures were performed under approved (A)BSL-2 conditions.

### 2.3. Centralized Gene Construction and Phylogenetic Analysis

The centralized consensus HA-con genes were created as previously described [11,14]. In brief, sequences that represent the major branches of the phylogenetic tree for each influenza subtype were selected. The representative sequences were aligned using ClustalX, and the most common amino acid at each position was determined as the consensus sequence. The alignments and analyses indicate that these centralized genes retain critical functional domains such as the secretory signal, polybasic connecting peptide, cleavage and fusion sites, transmembrane domains, and cytoplasmic tail. The consensus HAs, named H1-con and H3-con, were 566 amino acids in length. Phylogenetic analyses were performed using MacVector v.18.6.4 (MacVector, Inc., Apex, NC, USA) and Geneious Prime v11.1.5 (Dotmatics, Boston, MA, USA) softwares.

### 2.4. Expression of Plant-Made HA

Consensus HA-con sequences were codon-optimized for plant expression and synthesized by GeneScript, Inc. (Piscataway, NJ, USA). The mature coding region for each consensus sequence was genetically fused to a 5′ coding sequence corresponding to the *Nicotiana benthamiana* (Nb) extensin signal peptide such that the N-terminus of the resulting chimeric, recombinant protein would efficiently enter the plant endomembrane system for correct folding. Each gene was subcloned into transient virus expression vectors as previously described [32]. The resulting plant expression vectors containing the consensus H1-con (pKBP111) and H3-con (pKBP117) were used to transfect separate batches of wild-type Nb plants. After 7 days of incubation, transfected plants were harvested, and soluble proteins were extracted using 200 mM sodium phosphate and 250 mM sodium chloride, pH 9.0. Each extract was clarified using plate/frame filtration with diatomaceous earth filter aid. HA-con antigens were captured using TALON metal affinity chromatography (Cytiva, Thermo-Fisher Scientific, Waltham, MA, USA), with elution achieved by stepwise increase of imidazole concentration. Each TALON eluate was further purified using Capto-Q anion-exchange chromatography (Cytiva, Thermo-Fisher Scientific) and formulated by diafiltration into phosphate-buffered saline. The quality of each recombinant HA-con antigen was analyzed by BCA to determine protein concentration, SDS-PAGE for purity and analysis of degradation species, and molecular size consistency by size exclusion chromatography (SEC)-HPLC. Primary impurities observed were noted to be multimers of HA proteins under reducing SDS-PAGE conditions.

### 2.5. Vaccines and Immunizations

A well-characterized lysine-modified TMV was genetically engineered to express a single reactive N-terminal lysine (K) residue on the surface of the TMV virion [33]. All conjugation and vaccine preparation processes were performed as previously described [21]. In brief, HA-con protein was conjugated to TMV using EDC (1-Ethyl-3-[3-dimethylaminopropyl] carbodiimide hydrochloride) and Sulfo-NHS (*N*-hydroxysulfosuccinimide, Thermo-Fisher) reagents according to the manufacturer’s instructions [34]. Equal quantities of HA-con protein and TMV were mixed with 0.1 M MES pH 5.0 in 0.5 M NaCl. The mixture was added to freshly prepared EDC to 2 mM, immediately vortexed, and Sulfo-NHS was added to 5 mM. The mixture was vortexed at room temperature for 60 min. The reaction was stopped by adding hydroxylamine-HCl to 10 mM. The efficacy of conjugation was evaluated by SDS-PAGE and stained with Coomassie blue (R-250). Conjugation reactions were considered complete when no free HA protein was visible. Vaccines were prepared after conjugation by removal of reactive agents by dialysis against PBS in Slide-a-Lyzer cartridges (Thermo-Fisher Scientific) overnight. Protein quantity was determined by the bicinchoninic acid assay (BCA, BioRad, Hercules, CA, USA). The Fluzone vaccine consisted of a triton X-100 split virus vaccine suspended in sodium phosphate-buffered isotonic sodium chloride solution with 0.05% gelatin (NR-10483, Lot# U2248AA, BEI). The influenza viruses A/New Caledonia/20/99 (H1N1), A/Wisconsin/67/05 (H3N2), and B/Malaysia/2506/04 were grown in embryonated eggs and purified in a linear sucrose density gradient solution using continuous flow centrifuge. The standardized HA content was diluted into 0.25 mL at a ratio of 7.5 μg each of the HA from the influenza viruses.

In H1-HA groups, BALB/c mice (n = 50) were randomly allocated into 5 groups with 10 mice each. The groups of mice received a prime immunization with the following vaccines at day 0: TMV-H1-con + AddaVax, H1-con + AddaVax, Fluzone 2006–2007, AddaVax, and DPBS groups. The TMV-H1-con + AddaVax and H1-con + AddaVax groups were IM vaccinated with 100 μL (30 μg total HA protein) of TMV-H1-con and H1-con with adjuvant (AddaVax™, InvivoGen, San Diego, CA, USA). The Fluzone 2006–2007 group was IM vaccinated with 50 μL containing 4.5 μg HA proteins (9 μg total HA protein) of a Fluzone^®^ Trivalent Influenza Virus Vaccine, 2006–2007 Formula. The AddaVax and DPBS groups were IM vaccinated with 50 μL of AddaVax™ or DPBS, respectively, and served as controls. The mice were boosted at day 21 with the exact same vaccines as the prime immunization.

In H3-HA groups, BALB/c mice (n = 50) were randomly allocated into 5 groups with 10 mice each. The groups of mice received a prime immunization with the following vaccines at day 0: TMV-H3-con + AddaVax, H3-con + AddaVax, Fluzone 2006–2007, AddaVax, and DPBS groups. The TMV-H3-con + AddaVax and H3-con + AddaVax groups were IM vaccinated with 100 μL (30 μg total HA protein) of TMV-H3-con and H3-con with AddaVax™. The Fluzone 2006–2007 group was IM vaccinated with 50 μL containing 4.5 μg HA proteins (9 μg total HA protein) of Fluzone^®^ Trivalent Influenza Virus Vaccine, 2006–2007 Formula. The AddaVax and DPBS groups were IM vaccinated with 50 μL of AddaVax™ or DPBS, respectively, and served as controls. The mice were boosted at day 21 with the exact same vaccines as the prime immunization.

### 2.6. Immune Correlate Studies

One hundred female BALB/c mice were divided into two experiments with 50 mice each: H1-HA and H3-HA groups. All vaccinated groups were intramuscularly (IM) prime/boost immunized with different vaccines at 0- and 21-days post-vaccination (DPV) with a 27-gauge needle into both quadriceps in two 25 or 50 μL injections. All immunizations, bleeds, and tissue collections were performed under isoflurane or ketamine and xylazine-induced anesthesia (Figure 1).

Groups of mice were vaccinated on day 0 and boosted with the same vaccine as previously described on day 21. To determine immune correlates following vaccination, five mice from each group were sacrificed after prime (21 DPV) and boost (35 DPV) immunizations for collections of blood, spleens, and lungs. Blood was collected via cardiac puncture. Sera were separated from whole blood using a BD Microtainer Blood Collection Tube (Becton Dickenson, Becton, NJ, USA) and used for HI assay, microneutralization assay, and ELISA. Splenocytes were isolated by passing the spleen through a 40 μm Nylon Cell strainer (Fisher Scientific, Waltham, MA, USA), and red blood cells were lysed using ACK lysis buffer. Splenocytes were resuspended in cRPMI-1640 supplemented with 10% FBS and used for ELISpot assays. Lungs were homogenized in DPBS and centrifuged at 21,000× *g* for 10 min. Supernatants were collected and treated with 10 mM dithiothreitol (DTT) in DPBS at 37 °C for 1 h followed by 10% BSA in DPBS at 37 °C for 1 h before being used in ELISA.

### 2.7. Hemagglutination Inhibition (HI) Assay

Sera were incubated with receptor-destroying enzyme (RDE, 370013, Denka Seiken, Tokyo, Japan) overnight at 37 °C followed by inactivation at 56 °C for 30 min as described by the manufacturer’s protocol. Sera were serially diluted two-fold in DPBS in 96-well V-bottom plates. After diluting the sera, an equal volume (25 µL) of 4 HAU of the virus was added to each well. The plates were incubated at room temperature for 1 h, and then 50 µL of 0.5% chicken red blood cells was added. The hemagglutination patterns were read after 30 min.

### 2.8. Microneutralization Assay (MNA)

Sera were heat-inactivated at 56 °C for 30 min and then 2-fold serially diluted in sterile 96-well U-bottom plates before the addition of 100 TCID_50_ units of virus per well. After 1 h of incubation at 37 °C with 5% CO_2_, 100 µL of MDCK cells (2 × 10^5^ cells/mL) were added and incubated overnight at 37 °C with 5% CO_2_. The plates were then washed 3 times with 0.2 mL of DPBS before adding DMEM supplemented with 0.0002% TPCK-trypsin to each well. The plates were incubated at 37 °C for 3 days with 5% CO_2_. Fifty microliters of 0.5% chicken red blood cells were added to each well and the hemagglutination titers were determined after 30 min.

### 2.9. ELISA

Sera and lung supernatants were analyzed for the IgG-, IgM-, and IgA-specific immune responses by ELISA as previously described [35]. Recombinant HA proteins of influenza viruses were used including: H1-HA proteins [A/Puerto Rico/8/1934 (H1N1, NR-19240, Lot# 4003248, BEI), A/Brisbane/59/2007 (H1N1, NR-28607, Lot# 70048698, BEI), and A/California/07/2009 (H1N1, NR-42635, Lot# 62593466, BEI)]; H3-HA proteins: [A/Perth/16/2009 (H3N2, NR-49734, Lot# 70058355, BEI), A/Wisconsin/67/2005 (H3N2, NR-49237, Lot# 70052281, BEI), and A/New York/55/2004 (H3N2, NR-19241, Lot# 59311910, BEI)]. Secondary antibodies used for ELISA were HRP-conjugated antibodies: goat anti-mouse IgG H + L (Cat# 2987457, Lot# AP308P, Invitrogen, Waltham, MA, USA), goat anti-mouse IgM (Cat# 1021-05, Lot# A1620-PB33B, SouthernBiotech, Birmingham, AL, USA), and goat anti-mouse IgA (Cat#1040-05, Lot# 3021-Y892D, SouthernBiotech). Sera (diluted 1:1000) were used to detect IgG and IgM responses. Dithiothreitol (DDT)-treated lung supernatants (diluted 1:4) were used to detect IgA immune responses.

Immunolon 4 HBX microtiter 96-well plates (Thermo-Fisher Scientific) were coated with 150 ng/well of HA protein in bicarbonate/carbonate coating buffer overnight at 4 °C before blocking with 10% BSA in DPBS-T at room temperature for 2 h. Sera and lung supernatants (100 µL) were added to the plates in duplicate and incubated at room temperature for 2 h. After incubation and washing, plates were incubated with a 1:5000 dilution of secondary HRP-conjugated goat anti-mouse antibodies in 5% skim milk in DPBS-T at room temperature for 1 h. After further washing, plates were developed with 1-Step Ultra TMB-ELISA substrate (Thermo-Fisher Scientific). The reaction was stopped with 2 M sulfuric acid, and the optical density at 450 nm wavelength (OD_450_) was measured using a SpectraMax i3X multi-mode microplate reader (Molecular Devices, San Jose, CA, USA). The antibody titer was determined as a signal that was at least three times the background values.

### 2.10. ELISpot Assay

An interferon (IFN)-γ ELISpot assay was used to evaluate the T-cell response after vaccination. Total T cells were examined using pooled peptides from either H1-HA or H3-HA peptide arrays including: H1-HA peptides [A/Puerto Rico/8/1934 (H1N1, NR-18973, BEI) and A/Brisbane/59/2007 (H1N1, NR-18970, BEI)] and H3-HA peptides [A/Uruguay/716/2007 (H3N2, NR-18968, BEI) and A/Wisconsin/67e5/2005 (H3N2, NR-9472, BEI)].

Polyvinylidene 96-well difluoride-backed plates (MultiScreen-IP Filter plate, Sigma-Aldrich, St. Louis, MO, USA) were coated with 50 µL of anti-mouse IFN-γ monoclonal antibody (mAb) AN18 (5 µg/mL, Mabtech AB, Nacka Strand, Sweden) overnight at 4 °C. Plates were washed and blocked with cRPMI-1640 supplemented with 10% FBS for 1 h at 37 °C. A total of 100 µL of cell suspension (2.5 × 10^5^ cells) of mouse splenocytes was added to each well and stimulated with 50 µL of peptides (5 µg/mL/peptide) and incubated overnight at 37 °C with 5% CO_2_ to allow for IFN-γ production.

After incubation, plates were washed 6 times with DPBS and incubated with 50 µL/well of biotinylated anti-mouse IFN-γ R4-6A2 mAb (1:1000 dilution, Mabtech AB) diluted in DPBS with 1% FBS for 1 h at room temperature. Plates were then washed 6 times with DPBS and incubated with 50 µL/well of streptavidin–alkaline phosphatase conjugate (1:1000 dilution, Mabtech AB) diluted in DPBS with 1% FBS for 1 h at room temperature. After another 6 washes with DPBS, plates were developed by adding 100 µL of BCIP/NBT (Plus) alkaline phosphatase substrate (Thermo-Fisher Scientific). Development was stopped by washing several times with distilled water (dH_2_O). The plates were air-dried, and spots were counted using an automated ELISpot plate reader (AID iSpot Reader Spectrum, AID GmbH, Strasberg, Germany). Results were normalized with the negative control and expressed as spot-forming cells (SFCs) per 10^6^ splenocytes.

### 2.11. Influenza Virus Challenges in Mice

One hundred female BALB/c mice were divided into two experimental groups with 50 mice each: H1-HA and H3-HA groups. All groups were intramuscularly (IM) prime/boost immunized with different vaccines at 0- and 21-days post-vaccination (DPV), as mentioned above. At 35 DPV (0-days post-challenge, DPC), mice were intranasally challenged with 20 μL of virus inoculum containing 10 MLD_50_ of mouse-adapted influenza A viruses including A/Puerto Rico/8/1934 (H1N1), A/Fort Monmouth/1/1947 (H1N1), A/Texas/1/1977 (H3N1), and A/Aichi/2/1968 (H3N2). Mice were monitored daily for weight loss and euthanized when they lost 25% of their starting weight (Figure 2).

### 2.12. Statistical Analysis

GraphPad Prism 9 software was used to analyze all data. Data are expressed as the mean with standard error of the mean (SEM). One-way analysis of variance (ANOVA) with Tukey’s multiple comparisons test was used for analyzing data. The area under the curve of weight loss was calculated, and the multiple comparisons were performed using ANOVA. Survival outcomes were analyzed using the Kaplan–Meier log-rank test. A *p*-value < 0.05 was considered statistically significant.

## 3. Results

### 3.1. Sequence Analysis of HA Proteins

The centralized HA genes localized to the center of the phylogenetic tree relative to the wild-type genes, as designed [11,14]. The amino acid sequence alignment, similarities, and antigenic regions of the HA proteins are shown in Figure 3. The range of sequence identities in the wild-type H1 HA ranged from 79.3% to 86.4%, whereas the H1-con range was 82.7% to 92.2% (Figure 3A). The range of sequence identities in the wild-type H3 HA ranged from 86.7% to 98.9%, whereas the H3-con range was 90.8% to 94.5% (Figure 3B). The alignment shows the previously described antigenic sites (Sa, Sb, Ca, and Cb) of the H1 and (A, B, C, D, and E) antigenic sites of the H3 HA proteins. The antigenic sites are indicated by the boxed regions and highlight the differences between the wild-type and centralized consensus genes [36,37,38,39,40,41].

### 3.2. Expression and Purification of HA Antigen

Plants were infiltrated, and plant materials were harvested, extracted, and purified by affinity column. The protein purification profiles of expressed HA-con proteins were analyzed using reducing SDS-PAGE and SEC-HPLC analyses. The extraction/purification processes showed purity of expressed HA-con proteins at approximately 75 kDa in size (Appendix A). The purity of the HA-con protein, in terms of the major monomer species, was more than 76% and 54% for H1-con and H3-con proteins, respectively, as determined by integration and quantitation of molecular peaks. The conjugation reactions generated acceptable characteristics, defined as the absence of free HA-con proteins (Appendix A).

### 3.3. IgG Response Following Immunization

After prime and boost immunization, the HI and microneutralization titers were detected at lower levels (less than 1:10) in all vaccinated groups, regardless of the vaccines and viruses used. However, the antibody response against influenza HA proteins was detected by an ELISA following vaccination. No HA-specific antibody responses were detected in mice immunized with AddaVax or DPBS throughout the experiment.

H1-HA-specific IgG responses were detected after prime immunization and boost immunization. Following prime immunization, immunized mice demonstrated trends of lower IgG responses (OD_450 nm_ < 1.0). However, increased IgG titers were observed in mice immunized with TMV-H1-con + AddaVax and H1-con + AddaVax after boost immunization, significantly higher than those in mice immunized with Fluzone, AddaVax alone, or DPBS (Figure 4A–C). The TMV-H1-con + AddaVax and H1-con + AddaVax groups exhibited significantly higher IgG levels against all three H1-HA proteins compared to the other groups following both prime and boost immunizations (Figure 4A–C). Specifically, the TMV-H1-con + AddaVax group showed the highest IgG levels against A/Puerto Rico/8/1934 and A/California/7/2009 H1-HA proteins (Figure 4A,B), while no significant difference in IgG levels against A/Brisbane/59/2007 H1-HA protein was observed between the TMV-H1-con + AddaVax and H1-con + AddaVax groups (Figure 4C).

H3-HA-specific IgG responses were low after prime immunization (OD_450 nm_ < 1.0), with no statistically significant differences among groups. However, increased IgG levels were detected in the TMV-H3-HA group, which showed significantly higher levels than the AddaVax and DPBS groups after boost immunization (Figure 4D–F). Both the TMV-H3-con + AddaVax and H3-con + AddaVax groups exhibited significantly higher IgG levels against A/Perth/16/2009 and A/New York/55/2004 H3-HA proteins compared to the other groups (Figure 4D,F). Specifically, while the H3-con + AddaVax group had significantly lower IgG levels against A/Wisconsin/67/2006 H3-HA protein compared to the TMV-H3-con + AddaVax group, no difference in IgG levels was observed between the H3-con + AddaVax and Fluzone 2006–2007 groups (Figure 4E).

### 3.4. IgM Response Following Immunization

The IgM response was detected in sera with lower levels (OD_450 nm_ < 1.0) after prime and increased following boost immunization (Figure 5A–F). For the H1-HA-specific IgM responses, there was no difference in the IgM titers among groups following prime, but the TMV-H1-con + AddaVax group had significantly higher IgM titers than the other groups after boost immunizations, regardless of the protein used (Figure 5A–C). For the H3-HA-specific IgM responses, no significant differences were observed among groups following prime immunization, but differences were observed following boost immunization (Figure 5D–F). Immunized mice from the TMV-H3-con + AddaVax and H3-con + AddaVax groups had significantly higher IgM responses against A/Perth/16/2009 and A/New York/55/2004 H3-HA proteins compared to the other groups following boost immunization (Figure 5D,F). Additionally, the TMV-H3-con + AddaVax group showed significantly higher IgM titers against A/Wisconsin/67/2005 H3-HA protein compared to the other groups (Figure 5E).

### 3.5. IgA Response Following Immunization

The HA-specific IgA responses in lung supernatants were analyzed following prime and boost immunizations (Figure 6A–F). For H1-HA-specific IgA response, IgA was detected following prime immunization, though there were no significant differences (Figure 6A–C). After boost immunization, increased IgA responses were observed in mice immunized with TMV-H1-con + AddaVax and H1-con + AddaVax, although there were no significant differences between groups against A/Puerto Rico/8/1934 and A/Brisbane/59/2007 H1-HA proteins (Figure 6A,B). However, significantly higher IgA levels were observed in mice immunized with TMV-H1-con + AddVax and H1-con + AddaVax against A/California/7/2009 H1-HA protein (Figure 6C).

The induction of IgA responses against H3-HA proteins was relatively low following prime immunization and increased slightly following boost immunization (Figure 6D–F). The TMV-H3-con + AddaVax and H3-con + AddaVax groups had significantly higher IgA levels against A/Perth/16/2009 and A/New York/55/2004 H3-HA proteins than mice immunized with Fluzone, AddaVax alone, or DPBS following boost immunization (Figure 6D,F). Meanwhile, mice immunized with Fluzone had significantly higher IgA levels against A/Wisconsin/67/2005 H3-HA protein than the DPBS group (Figure 6E).

### 3.6. Cross-Reactive T-Cell Responses

The induction of cross-reactive T-cell response was measured by IFN-γ^+^ ELISpot using peptide libraries representing H1 and H3 strains (Figure 7A–D). We observed relatively lower IFN-γ^+^ SFC in all vaccinated groups, regardless of the peptide libraries used. As depicted in Figure 7A,B, one mouse in the TMV-H1-con + AddaVax group and two out of five mice in the Fluzone 2006–2007 group exhibited higher IFN-γ^+^ SFC against H1 peptide libraries. Similarly, only one mouse in the TMV-H3-con + AddaVax group demonstrated higher IFN-γ^+^ SFC (82.6 and 132.0 IFN-γ^+^ SFC/10^6^ splenocytes) against H3 peptide libraries (Figure 7C,D). However, there were no significant differences in spot numbers among the groups.

### 3.7. Protection against Historical Influenza Challenge

We evaluated the protective efficacy following vaccination against lethal challenge with 10 MLD_50_ of either historical H1 or H3 strains. All vaccinated groups provided varying degrees of protection against historical influenza strains, while the adjuvant- and DPBS-control groups succumbed to all lethal challenges by 7–8 days post-challenge (Appendix A).

For H1 influenza challenges (Figure 8, Appendix A), mice in the TMV-H1-con + AddaVax and Fluzone 2006–2007 groups exhibited slight weight loss (~10%) by day 7 after challenge with A/Puerto Rico/8/1934 (H1N1) before recovering without fatalities (Figure 8A,B). In contrast, immunized mice in the H1-con + AddaVax group experienced maximum weight loss (~20%) by day 8 after challenge with A/Puerto Rico/8/1934 (H1N1), with 80% of mice surviving; one mouse was euthanized on day 8 (Figure 8C). Upon challenge with A/Fort Monmouth/1/1947 (H1N1), mice in the TMV-H1-con + AddaVax and H1-con + AddaVax groups displayed increased weight loss (~15%) by day 7 before recovering (Figure 8D,E). All mice in the TMV-H1-con + AddaVax group survived after lethal challenge, while one mouse in the H1-con + AddaVax group succumbed to the infection. In contrast, four out of five mice in the Fluzone 2006–2007 group did not survive the lethal challenge (Figure 8F).

In the H3 influenza challenges (Figure 9, Appendix A), mice immunized with H3-con + AddaVax and Fluzone 2006–2007 showed slight weight loss without fatalities after challenge with H3N1 A/Texas/1/1977 (Figure 9A–C). However, all vaccinated groups exhibited weight loss by days 7–8 following challenge with H3N2 A/Aichi/2/1968 (Figure 9D,E). Notably, the H3-con + AddaVax group demonstrated 80% protection against H3N2 A/Aichi/2/1968 challenge, whereas immunized mice with Fluzone failed to protect against this virus, resulting in fatalities by day 9 (Figure 9F).

## 4. Discussion

A vaccine production platform that is rapid, cost-effective, and reliable is essential for controlling and preventing pandemic diseases. Plant-based nanoparticles conjugated with vaccine immunogens are widely adopted for mass vaccine production against various pathogens [42,43,44]. Plant-based systems facilitate crucial post-translational modification, such as glycosylation [45], which are pivotal for the functionality and stability of immunogens [46]. The main advantages of plant-based vaccine manufacturing include rapid production capability at large scales, economy of scale, and significantly low costs—typically 0.1–1% of the expenses associated with immunogen production compared to mammalian cell culture platforms [47,48,49,50,51,52]. These attributes make plant-based platforms highly suitable for meeting urgent global health needs during pandemics.

In this study, we generated centralized hemagglutinin (HA-con) immunogens designed to mimic the common ancestor of influenza viruses. These immunogens were genetically central to all other variants, aiming to enhance cross-protective immunity. Previous research has demonstrated that centralized HA immunogens elicit higher levels of cross-protection compared to wild-type immunogens [11]. Vaccinations utilizing centralized HA antigens have shown efficacy in providing cross-protection against multiple heterologous lethal influenza challenges. Specifically, when these immunogens were delivered via adenoviral vectors, they were compared favorably to seasonal influenza vaccines [11,12,14,53]. Despite demonstrating improved protective efficacy, the adenoviral vector vaccine platform faces significantly limitations due to safety concerns and potential interference from pre-existing immunity resulting from multiple prior immunizations [54,55].

TMV nanoparticles have proven effective as antigen delivery vectors for enhancing protective efficacy against influenza virus infections. Previous studies have shown that vaccination with TMV influenza immunogens induces robust immune responses that effectively reduce viral load, limit lung pathology, and result in low morbidity and mortality rates [21,56]. Importantly, pre-existing immunity to TMV does not diminish the ability to prime or boost immunity in mice [21]. Therefore, our objective was to develop a universal influenza vaccine using plant virus-based nanoparticles, specifically TMV, conjugated with centralized HA immunogens from influenza A viruses. In this study, we demonstrate that our vaccine elicits strong cross-reactive antibody responses against a range of historical influenza viruses following prime-boost immunization in mice. Furthermore, mice immunized with TMV-HA conjugated vaccines were effectively protected against infections from historical H1 influenza viruses, and showed partial protection against historical H3 influenza viruses.

The HI titer is commonly utilized to assess the efficacy of influenza vaccines, with an HI titer of at least 1:40 generally considered indicative of a 50% reduction in the risk of influenza infection in humans, often serving as a threshold [57,58,59]. Alternatively, the microneutralization (MN) assay evaluates the humoral immune response post-infection or vaccination by assessing serum antibody’s ability to prevent virus infection in vitro, reflecting antibody-mediated protection. In our study, HI titers, as well as MN titers, were detected at levels below 40 in all vaccinated groups, irrespective of the vaccines. While HI titers are widely used to gauge vaccine-induced responses, they have been criticized for their limited sensitivity and their mechanistic relevance to natural cellular infection. It is important to note that the viruses used in our HI and MN assays were propagated in embryonated eggs, which may not fully replicate natural infection responses. These serological assays may have limitations, particularly when the hemagglutination pattern is influenced by the type of erythrocytes used. Moreover, the viral HA exhibits species-specific differences in its ability to hemagglutinate erythrocytes, which can impact hemagglutination inhibition processes [60,61]. While neutralization is a critical function of antibodies, other Fc effector functions such as antibody-dependent cellular cytotoxicity (ADCC) and antibody-dependent phagocytosis (ADP) are also important for influenza protection [62,63]. These protective mechanisms warrant further investigation in our ongoing research.

Although we did not detect significant antibody responses using HI and MN assays, our results reveal robust induction of immunoglobulins against various HA proteins. Mice immunized with TMV-HA-conjugated and HA-con vaccines exhibited higher levels of antibody responses compared to those vaccinated with the commercial whole-inactivated influenza vaccine (Figure 4A–C, Figure 5A–C, and Figure 6A–C). Furthermore, mice vaccinated with TMV-H1-con showed comparable levels of protection against historical H1 influenza viruses, experiencing only slight weight loss without mortality (Figure 8, Appendix A). This protection from lethal H1 challenges correlates with the strong antibody responses observed in TMV-H1-con-immunized mice. These findings are consistent with previous research demonstrating that using centralized consensus HA immunogens can induce cross-immunity against diverse influenza strains [14].

Conversely, we observed that protection against lethal H3 influenza viruses in the H3-vaccinated groups was less effective compared to the H1-vaccinated groups. Despite high antibody responses observed following vaccination with TMV-H3-con and H3-con vaccines, mice immunized with TMV-H3-con succumbed to lethal H3 challenges, whereas those vaccinated with H3-con showed complete protection against H3N1 A/Texas/1/1977 and partial protection against H3N2 A/Aichi/2/1968 challenges (Figure 9, Appendix A). These results suggest that the efficacy of the TMV-H3-con vaccine may have been compromised due to impurity in the H3-con protein and/or the method used for conjugation with TMV. The conjugation method plays a critical role in the quality of the vaccine and the immune response it elicits [21]. Therefore, to optimize efficacy and maintain immunogenicity, it is essential to validate conjugation methods that ensure optimal interaction between TMV and HA proteins. These findings underline the importance of stringent quality control measures in vaccine development, particularly regarding the conjugation process, to maximize protective efficacy of influenza virus vaccine.

In general, vaccines are most effective against viruses with similar genetic identity. This is central to the consensus concept. However, small mutations in antigenic regions can have a significant impact on vaccine efficacy. In fact, single amino acid changes have been found to be responsible for substantial, or even complete, losses in vaccine efficacy [64,65,66]. Conversely, very small changes can significantly increase the breadth of immunity induced by a vaccine [67]. We compared the consensus H1 and H3 HA immunogens to the viruses that were used in this study and identified the major antigenic sites within the hemagglutinin proteins (Figure 3). The H1-Con gene was 91.3% and 95.2% identical to the challenge viruses PR/34 and FM/47, respectively. In this case, there were no significant differences in protective efficacy. In contrast, the Fluzone vaccine was only 88% and 91% identical to the PR/34 and FM/47 challenge strains, respectively. This may explain, in part, why there are observed differences in weight loss upon challenge, and supports the correlation between vaccine efficacy and genetic relationship. The effect of genetic distance with vaccine efficacy is further supported in H3 virus challenge studies where H3-con is more genetically similar to TX/77 and Aichi/68 than the commercial Fluzone vaccine. Specific mutations that effect vaccine efficacy are much more difficult to discern. Antigenic sites such as Cb and Sb in the H1 HA and sites A and B in H3 HA proteins show high levels of variability, indicating phenotypic plasticity that would account for the high levels of genetic diversity between the influenza viruses. Without mutational analyses, extensive immunogenicity, and protection studies, the individual contributions of the amino acid differences remain unknown. The effects of the individual and combined amino acid mutations are made even more complex by variabilities in epitope recognition by the highly variable MHC genes, an often overlooked variable in vaccine studies.

Future research should focus on optimizing the conjugation methods between TMV nanoparticles and HA immunogens to enhance vaccine efficacy against diverse influenza strains, particularly H3 variants. Investigating alternative vaccine delivery systems or adjuvants may also improve immune responses. Additionally, exploring the role of non-neutralizing antibody functions, such as ADCC and ADP, could provide further insights into enhancing vaccine-induced protection.

## 5. Conclusions

This study investigated plant-based nanoparticles conjugated with centralized HA-con immunogens as a potential platform for universal influenza vaccines. Utilizing TMV nanoparticles, we aimed to elicit cross-reactive antibody responses against historical influenza strains in mice. The findings demonstrated robust antibody production and significant protection against H1 influenza viruses. These findings provide valuable insights and underscore exciting opportunities for improvement through optimized vaccine delivery systems, further advancing vaccine development.

## 6. Patents

No domestic or international patent applications or intellectual property have been submitted or disclosed for the work described in this manuscript.

## Figures and Tables

**Figure 1 vaccines-12-01100-f001:**
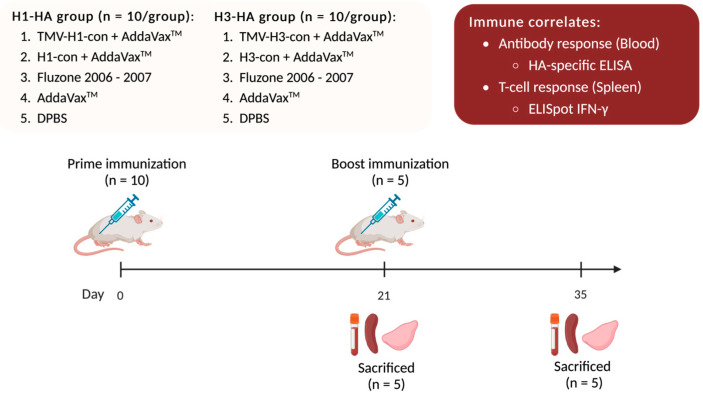
Immune correlates of the TMV-HA-con vaccines. BALB/c mice were divided into two experiments: H1-HA and H3-HA groups. All vaccinated groups were intramuscularly (IM) prime/boost immunized with different vaccines at 0- and 21-days post-vaccination (DPV). Following vaccination, five mice from each group were sacrificed at 21 and 35 DPV for collections of blood, spleens, and lungs. Blood was collected and sera were separated and used for hemagglutination inhibition (HI), microneutralization assays, and ELISA. Splenocytes were isolated for use in ELISpot assays. Lungs were homogenized, and lung supernatants were collected and used in HA-specific ELISA.

**Figure 2 vaccines-12-01100-f002:**
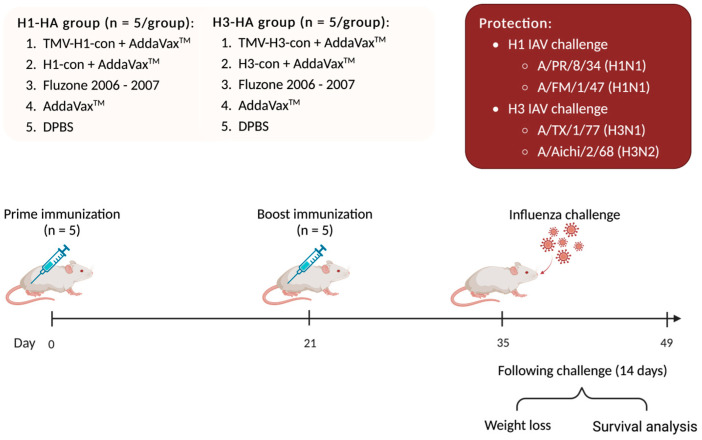
Protection against lethal challenge with historical influenza A viruses. BALB/c mice were divided into two experimental groups: H1-HA and H3-HA groups. All groups were intramuscularly (IM) prime/boost immunized with different vaccines at 0- and 21-days post-vaccination (DPV), as previously mentioned. At 35 DPV, mice were intranasally challenged with 20 µL of 10 MLD_50_ of mouse-adapted influenza A viruses including A/Puerto Rico/8/1934 (H1N1), A/Fort Monmouth/1/1947 (H1N1), A/Texas/1/1977 (H3N1), and A/Aichi/2/1968 (H3N2). Mice were monitored daily for weight loss for 14 days and euthanized when they lost 25% of their starting weight.

**Figure 3 vaccines-12-01100-f003:**
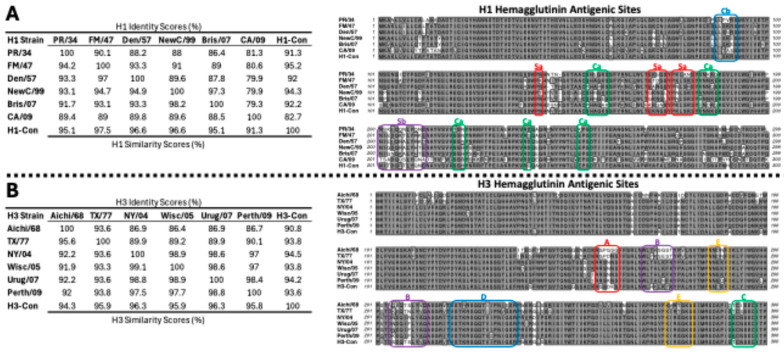
Sequence identity and Antigenic Site Characterization. All of the HA sequences used in the studies were aligned and analyzed for sequence identity and similarity scores and shown using a Gonnet similarity matrix. The H1-HA amino acid sequences were aligned using ClustalW, and the variable head domains and antigenic sites are indicated by the boxes (**A**). The H3-HA amino acid sequences were aligned using ClustalW, and the variable head domains and antigenic sites are indicated by the boxes (**B**). The similarity matrices and alignments were performed using MacVector (version 18.6.4). The (Sa, Sb, Ca, and Cb) of the H1 and (A, B, C, D, and E) antigenic sites of the H3 HA proteins are indicated by the boxed regions.

**Figure 4 vaccines-12-01100-f004:**
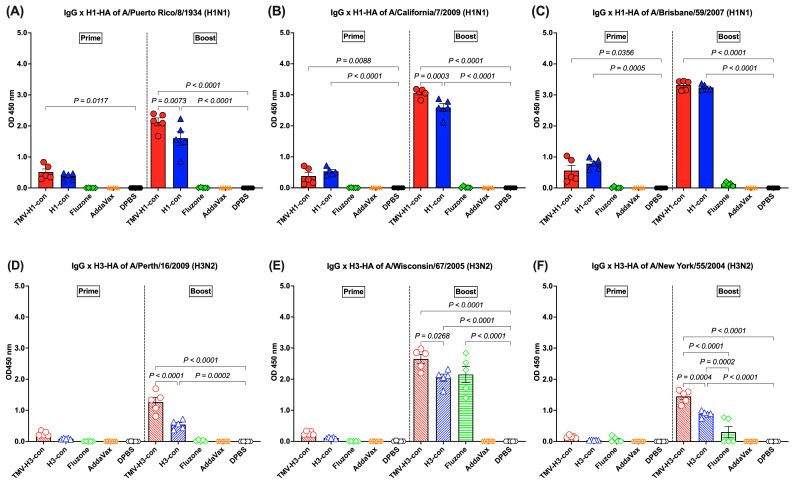
IgG responses in sera against panels of human H1-HA or H3-HA proteins after vaccination as measured by ELISA. Sera from immunized mice in the H1-HA groups were collected and tested with H1-HA proteins of (**A**) H1N1 A/Puerto Rico/8/1934, (**B**) H1N1 A/California/7/2009, and (**C**) H1N1 A/Brisbane/59/2007. Sera from immunized mice in the H3-HA groups were collected and tested with H3-HA proteins of (**D**) H3N2 A/Perth/16/2009, (**E**) H3N2 A/Wisconsin/67/2005, and (**F**) H3N2 A/New York/55/2004.

**Figure 5 vaccines-12-01100-f005:**
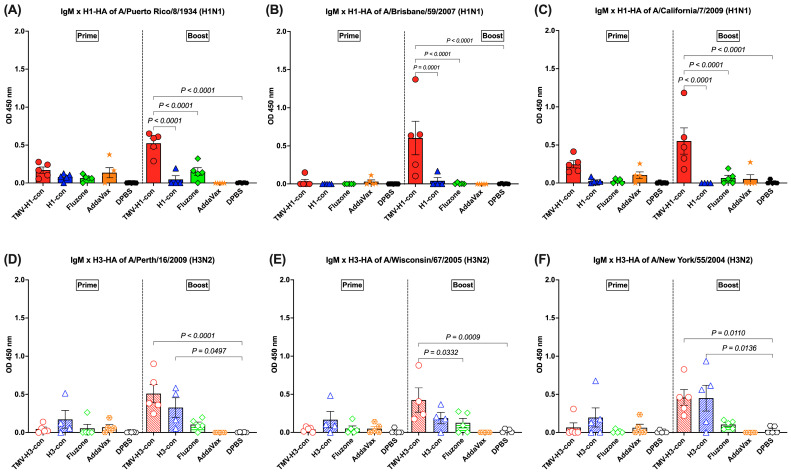
IgM responses in sera against panels of human H1-HA or H3-HA proteins after vaccination as measured by ELISA. Sera from immunized mice in the H1-HA groups were collected and tested with H1-HA proteins of (**A**) H1N1 A/Puerto Rico/8/1934, (**B**) H1N1 A/California/7/2009, and (**C**) H1N1 A/Brisbane/59/2007. Sera from immunized mice in the H3-HA groups were collected and tested with H3-HA proteins of (**D**) H3N2 A/Perth/16/2009, (**E**) H3N2 A/Wisconsin/67/2005, and (**F**) H3N2 A/New York/55/2004.

**Figure 6 vaccines-12-01100-f006:**
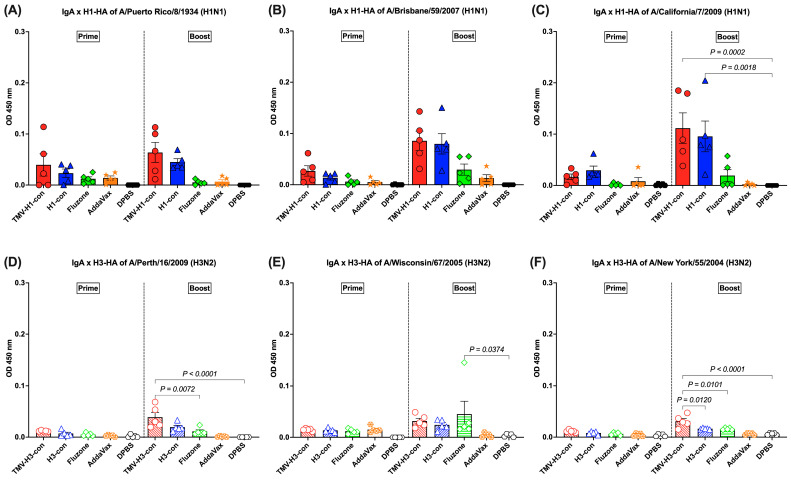
IgA responses in lung supernatants against panels of human H1-HA or H3-HA proteins after vaccination as measured by ELISA. DDT-treated lung supernatants from immunized mice in the H1-HA groups were tested with H1-HA proteins of (**A**) H1N1 A/Puerto Rico/8/1934, (**B**) H1N1 A/California/7/2009, and (**C**) H1N1 A/Brisbane/59/2007. DDT-treated lung supernatants from immunized mice in the H3-HA groups were tested with H3-HA proteins of (**D**) H3N2 A/Perth/16/2009, (**E**) H3N2 A/Wisconsin/67/2005, and (**F**) H3N2 A/New York/55/2004.

**Figure 7 vaccines-12-01100-f007:**
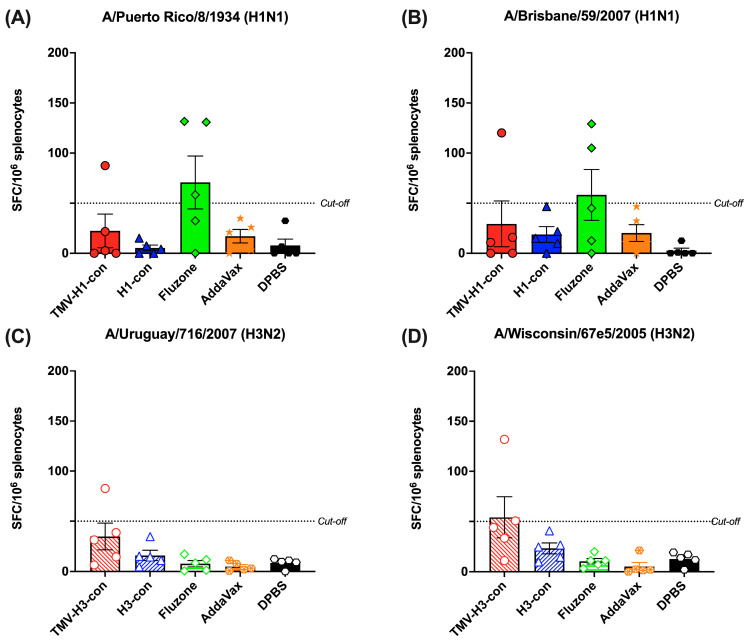
Total T-cell responses with different panels of human HA peptide libraries. Splenocytes from the H1-HA groups were isolated and analyzed using ELISpot IFN-γ with human H1 influenza peptide libraries: (**A**) H1N1 A/Puerto Rico/8/1934 and (**B**) H1N1 A/Brisbane/59/2007. Similarly, splenocytes from the H3-HA groups were isolated and analyzed with human H3 influenza peptides libraries: (**C**) H3N2 A/Uruguay/716/2007 and (**D**) H3N2 A/Wisconsin/67e5/2005. Dashed lines represent the cut-off values (≥50 SFC/10^6^ splenocytes).

**Figure 8 vaccines-12-01100-f008:**
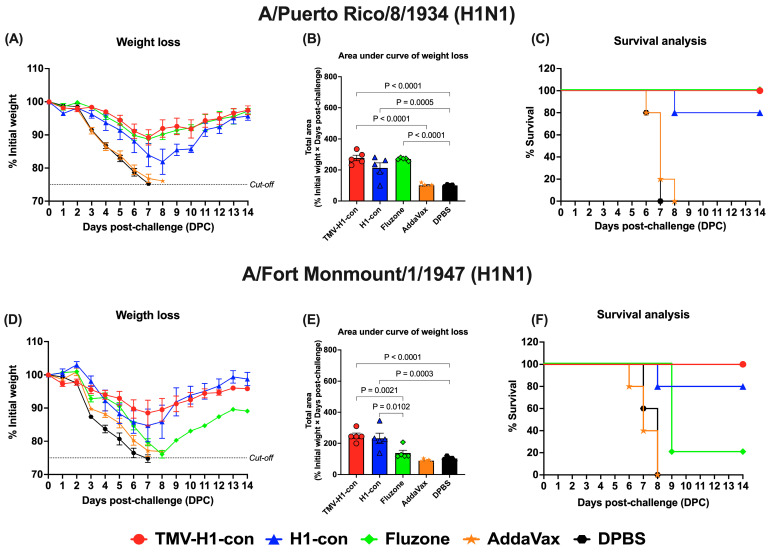
Protection against historical H1 influenza A viruses in mice. Mice were primed and boosted with immunization at days 0 and 21, respectively. Two weeks after boost immunization, mice were challenged intranasally with 10 MLD_50_ of H1N1 A/Puerto Rico/8/1934 (panels (**A**–**C**)) or H1N1 A/Fort Monmouth/1/1947 (panels (**D**–**F**)). The figures depict the percentage of weight loss and area under the curve monitored over 14 days post-challenge. Animals that exhibited 25% or more weight loss were humanely euthanized.

**Figure 9 vaccines-12-01100-f009:**
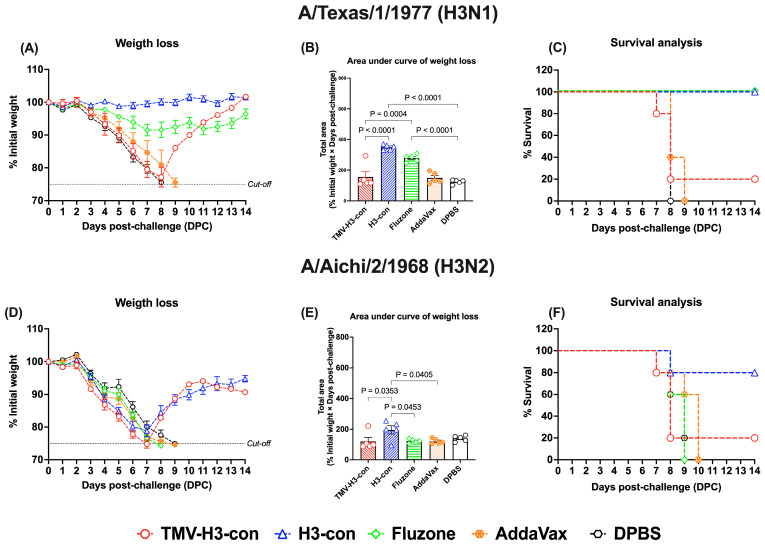
Protection against historical H3 influenza A viruses in mice. Mice were primed and boosted with immunization at days 0 and 21, respectively. Two weeks after boost immunization, mice were challenged intranasally with 10 MLD_50_ of H3N1 A/Texas/1/1977 (panels (**A**–**C**)) or H3N2 A/Aichi/2/1968 (panels (**D**–**F**)). The figures depict the percentage of weight loss and area under the curve monitored over 14 days post-challenge. Animals that exhibited 25% or more weight loss were humanely euthanized.

## Data Availability

All data relevant to the study are available in the main figures or the Appendix A.

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
