# Peer review of "Immunity and Protective Efficacy of a Plant-Based Tobacco Mosaic Virus-like Nanoparticle Vaccine against Influenza a Virus in Mice"

_vaccines, 2024, doi:10.3390/vaccines12101100_

Round 1
Reviewer 1 Report
Comments and Suggestions for Authors
In this manuscript, the authors investigated the efficacy of a plant-based Tobacco Mosaic Virus-like nanoparticle vaccine using a mouse infection model, and showed that the vaccine was effective against H1 influenza infection, but less effective against H3 influenza virus infection. Although this is a potentially interesting manuscript, there are concerns that need to be addressed for publication.
The nucleotide and amino acid sequences of H1 and H3 HAs used for vaccination are required. Furthermore, the authors should compare the sequences between vaccine HAs and challenge virus HAs (H1N1, H3N1, H3N2), and discuss the efficacy of vaccines from the viewpoint of antigenic sites on HA proteins (A, B, C, D, and E for H3 HA; Ca, Cb, Sa, and Sb for H1 HA) in order to address the quwstion: why the vaccine against H3 virus was less effective compared to H1.
The resolution of Fig. 3, 4, and 5 is not sufficient. The name of the construct is not clear.
Comments on the Quality of English LanguageN/A
Reviewer 2 Report
Comments and Suggestions for Authors
Madapong et al. prepared a novel influenza vaccine (TMV-HA-con) and evaluated the vaccine compared with HA-con protein vaccine and whole-inactivated virus vaccine (Fluzone) in their immune responses. The topic is interesting, but the manuscript has various issues. The explanation of structures of TMV-H1 (H3)-con, H1 (H3)-con, and Fluzone 2006-2007 is not enough. These vaccines should be described in more detail. Description of immunization schedules also is not enough. Authors described “All vaccinated groups were intramuscularly prime/boost immunized with different vaccines at 0- and 21-days post-vaccination.” (lines 161-163). However, it is difficult to figure out which vaccines were immunized at priming phase and boosting phase in each immunization group from text. Immunization schedules for each immunization group should be described clearly. In Figure 3, TMV-H1-con+AddaVax group was immunized with TMV-H1-con+AddaVax both at priming phase and boosting phase, wasn’t it? Letters in Figures 3-8 are too small to read. They should be enlarged to be able to read. In the manuscript, many abbreviations are spelled out many times. Abbreviations should be spelled out only they first appeared in text except for Abstract. Below are the examples. hemagglutinin (HA): lines 53, 74, 488, 566; hemagglutination inhibition (HI): lines 174, 199, 207, 512; hemagglutination units (HAU): lines 103, 211; tobacco mosaic virus (TMV): lines 63, 505; 567; antibody-dependent cellular cytotoxicity (ADCC): lines 528, 561, antibody-dependent phagocytosis (ADP): lines 528, 562. Format of References is not unified. It should be checked carefully before submission.
Round 2
Reviewer 1 Report
Comments and Suggestions for Authors
In the first round of reviewing, this Reviewer requested the authors to discuss the efficacy of vaccines from the viewpoint of antigenic sites on HA proteins to address the question: why the vaccine against H3 virus was less effective compared to H1. Although the authors described the sequence comparisons in the Result 3.1 and Figure 3, this Reviewer could not find any discussions in the Discussion section. Discuss it.
Author Response
Response to Reviewer (round 2)
Manuscript ID: vaccines-3154986
Title: Immunity and Protective E9icacy of a Plant-based Tobacco Mosaic Virus-Like
Nanoparticle Vaccine Against Influenza A Virus in Mice
Dear Reviewers,
Thank you for your insightful comments and suggestions. Please see the responses to your
comments below. You will find that we corrected the manuscript and included the inserted
text in our response in order to help you identify improvements. We thank the reviewers for
their time and effort. The revisions, suggestions, and critiques are underlined and our edits
and responses are italicized.
Reviewer 1 Comments:
Comment 1: In the first round of reviewing, this Reviewer requested the authors to discuss
the efficacy of vaccines from the viewpoint of antigenic sites on HA proteins to address the
question: why the vaccine against H3 virus was less effective compared to H1. Although the
authors described the sequence comparisons in the Result 3.1 and Figure 3, this Reviewer
could not find any discussions in the Discussion section. Discuss it.
Thank you for this suggestion. We have included further discussion on vaccine efficacy,
genetic relationship, and antigenic sites in the discussion (lines 623-645).
The discussion now includes the following paragraph “In general, vaccines are most effective against viruses with similar genetic identity. This is central to the consensus concept.
However, small mutations in antigenic regions can have a significant impact on vaccine
efficacy. In fact, single amino-acid changes have been found to be responsible for
substantial, or even complete, losses in vaccine efficacy [65-67]. Conversely, very small
changes can significantly increase the breadth of immunity induced by a vaccine [68]. We
compared the consensus H1 and H3 HA immunogens to the viruses that were used in this
study and identified the major antigenic sites within the hemagglutinin proteins (Figure 3).
The H1-Con gene was 91.3% and 95.2% identical to the challenge viruses PR/34 and FM/47,
respectively. In this case there were no significant differences in protective efficacy. In
contrast, the Fluzone vaccine was only 88% and 91% identical to the PR/34 and FM/47
challenge strains. This may explain, in part, why there are observed differences in weight loss
upon challenge and supports the correlation between vaccine efficacy and genetic
relationship. The effect of genetic distance with vaccine e8icacy is further supported in the
H3 virus challenge studies where the H3-con is more genetically similar to TX/77 and
Aichi/68 than the commercial Fluzone vaccine. Specific mutations that affect vaccine
efficacy are much more difficult to discern. Antigenic sites such as Cb and Sb in the H1 HA
and sites A and B in H3 HA proteins show high levels of variability, indicating phenotypic
plasticity that would account for the high levels of genetic diversity between the influenza
viruses. Without mutational analyses, extensive immunogenicity, and protection studies, the
individual contributions of the amino acid differences remain unknown. The effects of the
individual and combined amino acid mutations are even more complexed by variabilities in
epitope recognition by the highly variable MHC genes, an often over-looked variable in
vaccine studies.”
Reviewer 2 Report
Comments and Suggestions for Authors
The manuscript was improved after revision. But the manuscript has still many careless mistakes. Authors should check manuscript more carefully before submission.
Letters in all figures are still too small to read. They should be larger. Especially, letters in Figure 3, Letters of p-value in Figures 3 to 9 are hard to read.
Comments on the Quality of English LanguageThere are still careless mistakes through manuscript. Improve them. Some examples are shown below.
Line 48. CDC stands for “Centers for Disease Control and Prevention”.
Line 116. Add names of companies for MacVector and Geneious (Gneious Prime?) software.
Line 130. Add name of city for the company if possible.
Lines 133, 154. “MA, USA” should be removed here (see line 131).
Line 165. “2006-2007” should be “Fluzone 2006-2007”.
Line 168. Describe name and location of company when the reagent (AddaVax) first appeared (line 165). The City name where company is located should be added if possible.
Line 207. Add name of city where company is located.
Line 228. Add a period after “CO2”.
Line 244. Add name of city where company is located.
Line 245. “AL, USA” should be removed here (see line 244).
Line 268. Add name of city where company is located.
Line 279. Remove “Nacka Strand, Sweden” here. The location of company should be written only when it appeared first.
Line 318. Use “amino acid sequence” instead of “sequence”.
Line 330. Use “amino acid sequence” instead of “protein sequence”.
Line 334. “The (Sa, Sb, Ca, and Cb) of the H1 and (A, B, C, D, and E) antigenic sites of the H3 HA proteins” is weird description. Correct it.
Lines 348, 380, 388, 405, 413, 433. Descriptions of “Immunoglobulin G (IgG)”, “Immunoglobulin M (IgM)”, “Immunoglobulin A (IgA)” are not necessary. Just “IgG”, “IgM”, “IgA” are enough.
Author Response
Response to Reviewer (round 2)
Manuscript ID: vaccines-3154986
Title: Immunity and Protective E9icacy of a Plant-based Tobacco Mosaic Virus-Like
Nanoparticle Vaccine Against Influenza A Virus in Mice
Dear Reviewers,
Thank you for your insightful comments and suggestions. Please see the responses to your
comments below. You will find that we corrected the manuscript and included the inserted
text in our response in order to help you identify improvements. We thank the reviewers for
their time and effort. The revisions, suggestions, and critiques are underlined and our edits
and responses are italicized.
Reviewer 2 Comments
Comments 1: The manuscript was improved after revision. But the manuscript has still many
careless mistakes. Authors should check manuscript more carefully before submission.
Response 1: Thank you for pointing this out. The details were revised and edited in the revised
manuscript according to the reviewer’s comments.
Comments 2: Letters in all figures are still too small to read. They should be larger. Especially,
letters in Figure 3, Letters of p-value in Figures 3 to 9 are hard to read.
Response 2: We agree with this comment. Therefore, we increased the front sizes in all
figures to make it easy to read as much as possible, especially letters of p-value in Figures 3-
9. In addition, Figure 3 was reconstructed in order to increase the size for improved
readability and clarity.
Point 1: Line 48. CDC stands for “Centers for Disease Control and Prevention”.
Response 1: The “Centers for Disease Control and Prevention” was corrected in the revised
manuscript (see lines 48-49).
Point 2: Line 116. Add names of companies for MacVector and Geneious (Gneious Prime?)
software.
Response 2: Names of companies for MacVector and Geneious softwares were added in the
revised manuscript (see lines 116-117).
Point 3: Line 130. Add name of city for the company if possible.
Response 3: City name “Waltham” was added in the revised manuscript (see line 132).
Point 4: Lines 133, 154. “MA, USA” should be removed here (see line 131).
Response 4: The “MA, USA” was removed in the revised manuscript (see lines 134 and 155)
Point 5: Line 165. “2006-2007” should be “Fluzone 2006-2007”.
Response 5: The “2006-2007” was corrected and replaced with “Fluzone 2006-2007” in the
revised manuscript (see line 165).
Point 6: Line 168. Describe name and location of company when the reagent (AddaVax) first
appeared (line 165). The city name where company is located should be added if possible.
Response 6: The city name of AddaVax was added in the revised manuscript (see line 168).
All the city names where companies are located were added after first appeared in the
revised manuscript (see lines 87, 121, 156, 168, 211, 247, and 260). Also, the locations of the
company were removed after first statement throughout the revised manuscript.
Point 7: Line 207. Add name of city where company is located.
Response 7: The city name of the company was added in the revised manuscript (see line
208)
Point 8: Line 228. Add a period after “CO2”.
Response 8: A period (.) was added after CO2 in the revised manuscript (see line 229).
Point 9: Line 244. Add name of city where company is located.
Response 9: The city names of the companies were added in the revised manuscript (see
line 244).
Point 10: Line 245. “AL, USA” should be removed here (see line 244).
Response 10: The “AL, USA” was removed in the revised manuscript (see line 248).
Point 11: Line 268. Add name of city where company is located.
Response 11: The city name of the company was added in the revised manuscript (see line
272)
Point 12: Line 279. Remove “Nacka Strand, Sweden” here. The location of company should
be written only when it appeared first.
Response 12: The “Necka Strand, Sweden” was removed in the revised manuscript (see lines
279 and 282).
Point 13: Line 318. Use “amino acid sequence” instead of “sequence”.
Response 13: The “amino acid sequence” was replaced instead of “sequence” in the revised
manuscript (see line 321).
Point 14: Line 330. Use “amino acid sequence” instead of “protein sequence”.
Response 14: The “amino acid sequence” was replaced instead of “protein sequence” in the
revised manuscript (see lines 333-334).
Point 15: Line 334. “The (Sa, Sb, Ca, and Cb) of the H1 and (A, B, C, D, and E) antigenic sites
of the H3 HA proteins” is weird description. Correct it.
Response 15: The descriptions of antigenic sites were changed in the revised manuscript.
Point 16: Lines 348, 380, 388, 405, 413, 433. Descriptions of “Immunoglobulin G (IgG)”,
“Immunoglobulin M (IgM)”, “Immunoglobulin A (IgA)” are not necessary. Just “IgG”, “IgM”,
“IgA” are enough.
Response 16: The “Immunoglobulin G (IgG), Immunoglobulin M (IgM) and Immunoglobulin A
(IgA)” were shortened into “IgG, IgM and IgA” in the revised manuscript (see lines 351, 383,
390, 407, 414, and 434)